# Combination of Recombinant Proteins S1/N and RBD/N as Potential Vaccine Candidates

**DOI:** 10.3390/vaccines11040864

**Published:** 2023-04-18

**Authors:** Noe Juvenal Mendoza-Ramírez, Julio García-Cordero, Sandra Paola Martínez-Frías, Daniela Roa-Velázquez, Rosendo Luria-Pérez, José Bustos-Arriaga, Jesús Hernández-Lopez, Carlos Cabello-Gutiérrez, Joaquín Alejandro Zúñiga-Ramos, Edgar Morales-Ríos, Sonia Mayra Pérez-Tapia, Martha Espinosa-Cantellano, Leticia Cedillo-Barrón

**Affiliations:** 1Departamento de Biomedicina Molecular, Cinvestav, Av. IPN # 2508 Col, Mexico City 07360, Mexico; 2Departamento de Bioquímica, Cinvestav, Av. IPN # 2508 Col, Mexico City 07360, Mexico; 3Unidad de Investigación en Enfermedades Oncológicas, Hospital Infantil de México Federico Gómez, Mexico City 06720, Mexico; 4Unidad de Biomedicina, Facultad de Estudios Superiores-Iztacala, Universidad Nacional Autónoma de México, Av. De los Barrios # 1, Col. Los Reyes Iztacala, Tlalnepantla 54090, Mexico; 5Laboratorio de Inmunología, Centro de Investigación en Alimentación y Desarrollo A. C (CIAD) Carretera a la Victoria km 0.6, Hermosillo Sonora 83304, Mexico; 6Instituto Nacional de Enfermedades Respiratorias Ismael Cosío Villegas (INER), Departamento de Investigación en Virología y Micología, Calzada de Tlalpan 4502, Belisario Domínguez, Tlalpan 14080, Mexico; 7Instituto Nacional de Enfermedades Respiratorias Ismael Cosío Villegas y Tecnologico de Monterrey, Escuela de Medicina y Ciencias de la Salud, Monterrey 64849, Mexico; 8Unidad de Desarrollo e Investigación en Bioterapéuticos (UDIBI), Escuela Nacional de Ciencias Biológicas, Instituto Politécnico Nacional, México City 11340, Mexico; 9Departamento de Infectómica y Patogénesis Molecular, Cinvestav, Av. IPN # 2508 Col, San Pedro Zacatenco, México City 07360, Mexico

**Keywords:** SARS-CoV-2, vaccine, nucleocapsid protein, spike protein, RBD domain

## Abstract

Despite all successful efforts to develop a COVID-19 vaccine, the need to evaluate alternative antigens to produce next-generation vaccines is imperative to target emerging variants. Thus, the second generation of COVID-19 vaccines employ more than one antigen from severe acute respiratory syndrome coronavirus 2 (SARS-CoV-2) to induce an effective and lasting immune response. Here, we analyzed the combination of two SARS-CoV-2 viral antigens that could elicit a more durable immune response in both T- and B-cells. The nucleocapsid (N) protein, Spike protein S1 domain, and receptor binding domain (RBD) of the SARS-CoV-2 spike surface glycoproteins were expressed and purified in a mammalian expression system, taking into consideration the posttranscriptional modifications and structural characteristics. The immunogenicity of these combined proteins was evaluated in a murine model. Immunization combining S1 or RBD with the N protein induced higher levels of IgG antibodies, increased the percentage of neutralization, and elevated the production of cytokines TNF-α, IFN-γ, and IL-2 compared to the administration of a single antigen. Furthermore, sera from immunized mice recognized alpha and beta variants of SARS-CoV-2, which supports ongoing clinical results on partial protection in vaccinated populations, despite mutations. This study identifies potential antigens for second-generation COVID-19 vaccines.

## 1. Introduction

Severe acute respiratory syndrome coronavirus 2 (SARS-CoV-2) is the etiological agent of coronavirus disease 2019 (COVID-19), which has become one of the biggest threats to global health and has been responsible for over 6.7 million deaths worldwide since its emergence in November 2019 [1,2,3].

SARS-CoV-2 is an enveloped, positive-sense, single-stranded RNA virus that belongs to the coronavirus family. The SARS-CoV-2 genome encodes four structural proteins, membrane (M), envelope (E), spike (S), and nucleocapsid (N) [4,5], along with 13 nonstructural proteins. Cumulative data on COVID-19 have demonstrated that the S protein from the surface of SARS-CoV-2 is one of the most immunogenic proteins and has been used for the diagnosis and evaluation of the protective response [6,7]. Protein S is a highly glycosylated trimeric protein. This protein is organized into two subunits, S1 and S2 [8]. The S1 subunit binds to host receptors and S2 facilitates fusion between the viral envelope and the host cell membrane [9,10]. The S1 subunit contains the receptor-binding domain (RBD), a domain that allows interaction with the angiotensin-converting enzyme 2 (ACE2) receptor [11]. The transmembrane domain of the S2 subunit comprises heptapeptide repeat regions and fusion peptides. When it is hydrolyzed by host cell-derived serine protease TMPRSS2, it mediates fusion of the virus and host cell membrane through extensive conformational rearrangement [12]. The SARS-CoV-2 N protein is a highly conserved structural protein involved in RNA packaging and viral particle release [13,14].

Since the publication of the genomic sequence of SARS-CoV-2 in January 2020, global efforts have focused on the development of vaccines, many of which have been generated, and their use has succeeded in controlling the spread of the virus and saving millions of lives, although the distribution and coverage of these vaccines have not been equitable. Unfortunately, some vaccines currently in use employ cutting-edge technologies that cannot be easily deployed in developing countries [15]. It is therefore important to find alternatives to develop safe, effective, and accessible vaccines for these populations [16]. In addition, the emergence of new variants of SARS-CoV-2 has rendered some of the vaccines currently in use to be ineffective. To date, numerous mutations have been reported in the S protein of the SARS-CoV-2 original isolate, resulting in several variants of the virus [17]. As a result, vaccinated individuals have lower neutralizing antibodies for the emerging SARS-CoV-2 variants, including the Omicron variant [18,19,20].

Most SARS-CoV-2 vaccines currently in development deliver only the S antigen, as antibodies raised against the S antigen or RBD are expected to neutralize the infection [21,22]. Therefore, the use of SARS-CoV-2 antigens that induce an immune response independent of neutralization has been proposed [23,24,25,26]. Improved approaches should be implemented to guarantee the coverage of new variants or antigens that may induce a more protective response. Thus, the evaluation of different SARS-CoV-2 antigens is necessary for preclinical studies in animal models to identify new vaccine candidates that induce an effective and durable immune response.

Previous studies of other members of the coronavirus family have suggested that the N protein is highly immunogenic and a well-conserved sequence among different coronaviruses, making it an attractive alternative to be evaluated as a potential and effective target in subunit protein vaccination [27,28,29,30,31]. Some studies have evaluated the ability of the coronavirus N protein to elicit a sustained cellular immune response [32].

This study aimed to analyze whether a durable, humoral, and cellular immune response can be elicited through the combination of the N protein with a recombinantly designed, prepared, and purified S1 and RBD domain of the spike surface glycoprotein of SARS-CoV-2 in a mammalian expression system, which entails posttranslational modifications similar to those produced in the natural host, thereby avoiding immunogenicity incidents.

## 2. Materials and Methods

### 2.1. Expression and Purification of SARS-CoV-2 Proteins

#### 2.1.1. Eukaryotic System

The sequences encoding the S1 domain (nucleotides 21563-23738) and the N proteins (nucleotides 28274-29533) were based on the SARS-CoV-2 Wuhan isolate (GenBank MN908947.3) and were used to transform pcDNA3.1 plasmids, as reported by Garcia Cordero et al. [33]. The plasmid encoding the spike RBD (pCAGGS/RBD) was kindly provided by Prof. Florian Krammer (Icahn School of Medicine at Mount Sinai, New York, NY, USA).

The 3 recombinant proteins were expressed in Expi293 cells (Thermo Fisher Cat #A14527) and purified by Immobilized Metal-Affinity Chromatography (IMAC) and gel filtration chromatography as previously described [33].

#### 2.1.2. Procaryotic Expression System of Variants of Concern

Wuhan RBD was expressed in *Escherichia coli* (*E.coli*) SoluBL21 (Genlantis, San Diego, CA, USA), purified and refolded as previously described [34]. Based on the *E. coli* codon-optimized receptor binding domain (RBD) protein sequence from the spike protein (UniProt accession: P0DTC2) and using the methodology described above, RBD proteins of the alpha and beta variants of concern (VOCs) were expressed and purified.

### 2.2. Mouse Immunization Protocol

Groups of 6- to 8-week-old female BALB/c (H-2d) mice were obtained from the Laboratory of Animal Production and Experimentation Unit (UPEAL) at CINVESTAV. Protocol procedures (02-11-16) were approved by the Animal Use Ethical Committee, CINVESTAV, and all animals were handled in accordance with institutional guidelines. For antigen formulation, 10 µg of each SARS-CoV-2 protein was diluted in PBS and mixed with Freund’s adjuvant. Mice were divided into six groups, each containing 10 mice, and treated as follows: Group 1, protein N; Group 2, protein S1; Group 3, protein RBD; Group 4, proteins S1+N; Group 5, proteins RBD+N; and Group 6, vehicle.

Immunization was carried out on day 0, and booster shots were applied on days 20 and 60. Blood samples were collected on day 0 (first dose) and every 20 days thereafter (20, 40, 60 and 80 days). Serum was collected from blood and stored at −20 °C until assayed for antibodies against S1, RBD, and N proteins by ELISA, immunofluorescence, and determination of neutralizing antibodies. All samples were stored at −20 °C until analysis. The Animal Use Ethical Committee at CINVESTAV approved the protocols and procedures (number 02-11-16).

### 2.3. Immunofluorescence Assay

Vero cells were plated at a cell density of 6 × 10^4^ cells/well in a 24-well plate. Cells were infected with SARS-CoV-2 at an MOI of 1 in a Biosafety Laboratory Level 3 (BSL-3, Instituto Nacional de Enfermedades Respiratorias, Mexico city, Mexico). Cells were incubated for 24 h with 5% CO_2_ at 37 °C, washed with 1X PBS, and fixed with 4% paraformaldehyde (Sigma-Aldrich, St. Louis, MO, USA) for 30 min. Fixed cells were permeabilized with 0.1% Triton-X100 in PBS (PBS-Tr) for 30 min, blocked with 10% goat serum in PBS-Tr for 30 min, and incubated for 60 min with serum samples from immunized mice diluted (1:200) in PBS-Tri containing 10% goat serum. After washing three times with PBS/Trit, Cy3-conjugated goat anti-mouse IgG (A10521, Invitrogen) was added as a secondary antibody (1:500 dilution), incubated for 1 h, and washed 5 times. Finally, nuclei were labeled with DAPI/Vectashield (H-1200, Vector Labs, Burlingame, CA, USA). Images were taken using a confocal microscope.

### 2.4. Enzyme-Linked Immunosorbent Assay (ELISA)

ELISAs were performed in microplates (655061; Fisher Scientific, Carlsbad, CA, USA) coated overnight at 4 °C with 2 µg/mL (50 µL/well) of S1, N, and RBD proteins diluted in PBS 1X pH 7.4. For variants, microplates were coated overnight at 4 °C with 2 µg RBD proteins based in Wuhan isolate, Alpha variant, and Beta variant diluted in PBS 1X pH 7.4. RBD expressed in Expi293 cells was used as a control.

Plates were washed 3 times with PBS/0.1% Tween 20 and then blocked with PBS containing 5% (*w/v*) skim milk for 1 h. Next, serum diluted 1:100 in PBS containing 5% (*w/v*) skim milk was added, and plates were incubated for 2 h. After washing, the secondary antibodies goat anti-mouse IgM-HRP (G21040, Invitrogen) and goat anti-mouse IgG-HRP (62-6820, Invitrogen) were added at a dilution of 1:5000. For isotype determination, the secondary antibodies rabbit anti-mouse IgG1-HRP (61-0120, Invitrogen) and rabbit anti-mouse IgG2a-HRP (61-0220, Invitrogen) were added at a dilution of 1:2000. Plates were incubated for 1 h at 37 °C. After washing five times, bound conjugates were visualized using o-phenylenediamine (P9029; Sigma-Aldrich, St. Louis, MO, USA) and H_2_O_2_ (Sigma) as substrates, allowing the reaction to proceed for 15 min. The reaction was stopped with 50 μL of 2 N H_2_SO_4_ Optical density was measured at 450 nm using an ELISA Lektor (Multiskan FC, Thermo Scientific, Whaltan, USA)

### 2.5. Simulated Neutralization Assay

The simulated neutralization assay was performed as described by Abe et al. [35] with minor modifications. A Maxisorp ELISA microplate (Thermo Fisher Scientific, Waltham, MA, USA) was coated for 12 h at 4 °C with RBD (2 μg/mL) diluted in 50 mM carbonate buffer at pH 9.6. The plates were washed with PBS and blocked with 0.05% PBS-Tween, 2% BSA, and 3% glucose. Thereafter, the plates were incubated at room temperature for 1 h and washed with PBS. Serial dilutions of the serum were prepared in 1% PBS milk, and 50 μL of the dilution was added to the wells and incubated for 30 min at 20 °C. Four washes were performed with 0.05% PBS-Tween. Biotinylated ACE2 diluted in PBS (2 μg/mL) was added to the plates and incubated for 30 min. After washing, HRP-streptavidin (1:2000) was added and incubated for 30 min. After washing four times, TMB substrate solution (Sigma-Aldrich, St. Louis, MO, USA) was added and incubated for 20 min at 20 °C. The reaction was stopped with 50 μL of 1 M H_2_SO_4_. ACE2 was used in duplicate as a control. The results are expressed as % neutralization = (optical density problem well/optical density control well) × 100. Samples with values > 35 were considered positive for neutralizing antibodies.

### 2.6. Cytokine Assays

Mice were euthanized 120 days after the first immunization, and splenocytes were isolated. Splenocytes were cultured at 2 × 10^5^ density in 200 µL RPMI medium (5% FBS, 2 mM L-glutamine, 1X NEAA, 1X vitamins, 1X antibiotic/antimitotic) and stimulated with recombinant S1, N, and RBD proteins (2 μg/mL) for 48 h at 37 °C. Supernatants were collected, centrifuged at 200× *g*, and stored at −20 °C until evaluation. The cytokines TNF-α, IFN-γ, IL-2, IL-4, and IL-5 were quantified according to the instructions of the Cytometric Bead Array (CBA) Mouse Th1/Th2 Cytokine using the Cytometric Bead Array (CBA) Mouse Th1/Th2 Cytokine (551287, BD Biosciences, Franklin, NJ, USA) kit. Data were collected using a Cytometer BD FACS Canto II.

### 2.7. Prediction of Structure Protein

The structure of N, S1, and RBD proteins was predicted by AlphaFold server. The parameters “novo” prediction and minimization of energy were applied. The visualization and analysis of structures were performed using Chimera 1.15 software Appendix A.

### 2.8. Statistical Analyses

Nonparametric ANOVA and the Wilcoxon test in Prism v.8 software (GraphPad software, San Diego, CA, USA) were used for statistical analyses. Bars represent the mean ± SD. * *p* < 0.05, ** *p* < 0.01, *** *p* < 0.001. *p* ≤ 0.05 was considered statistically significant.

## 3. Results

### 3.1. Purification and Expression of Recombinant Proteins

As described previously [33], plasmids encoding N, S1, and RBD proteins (pcDNA3.1/S1, pcDNA3.1/N, and pCAGGS/RBD, respectively) were used to transfect mammalian Expi293 cells (Figure 1A). Purified proteins were analyzed using SDS-PAGE and Western blotting to detect recombinant proteins using anti-His antibody, as shown in Figure 1B. The proteins of interest corresponded to the predicted size of 120 kDa, 35 kDa, and 55 kDa for S1, RBD, and N proteins, respectively.

No reactivity was observed in Expi293 cells transfected with the parental plasmid using the same antibody. Purified proteins (Figure 1B) were used to immunize groups of BALB/c mice. Serum was collected at 20-day intervals for the determination of IgM and IgG antibodies using ELISA (Figure 1C).

### 3.2. SARS-CoV-2 Viral Proteins Are Recognized by Serum Samples in Infected Vero Cells

After the immunization was completed, we assessed whether the serum samples obtained from the mice immunized with recombinant proteins were able to recognize the native proteins produced by SARS-CoV-2. This was performed using Vero cells infected with a SARS-CoV-2 isolate (performed in the BSL-3 laboratory). A serum pool from different groups of mice immunized with each protein was used as the primary antibody. Recognition of native proteins and their characteristic distribution around the nuclei (N protein) or throughout the cytoplasm (S1 and RBD) of the infected cells were observed (Figure 2A). Preimmune sera and sera from mice immunized with vehicle alone did not show any recognition in infected cells (Figure 2B,C). Additionally, pooled immune sera showed no recognition in uninfected Vero cells (Figure 2D).

### 3.3. Humoral Response in Immunized Mice

Six groups of BALB/c mice were subjected to the immunization, as depicted in Figure 1C. After 20 days of each immunization with recombinant proteins, mice were bled, and serum samples were obtained to evaluate IgM- and IgG-specific antibody responses (Figure 3).

As expected, IgM antibodies peaked after 20 or 40 days and subsequently decreased. The group of mice immunized with N protein alone or in combination with either S1 or RBD (S1+N or RBD+N) elicited a low but specific IgM antibody response against the N protein. Low quantities of IgM-specific antibodies against the native S1 protein were detected in the groups immunized with S1, RBD, S1+N, and RBD+N, reaching a peak on day 20 and decreasing afterward. Compared to anti-S1 antibodies, lower levels of anti-RBD-specific IgM antibodies were detected in the S1, RBD, S1+N, and RBD+N groups, showing similar kinetics over time (Figure 3A–C). IgM antibody differences between immunized mice with proteins vs. mice treated just with vehicle were statistically significant (*** *p* < 0.001).

IgG antibody levels were considerably higher than IgM levels and increased steadily in the different groups as time progressed (Figure 3D–F). Specific IgG antibodies were detected against N (immunization groups N, S1+N, and RBD+N) as well as S1 and RBD (immunization groups S1, RBD, S1+N, and RBD+N). IgG differences between immunized mice with proteins vs. mice treated just with vehicle were statistically significant (*** *p* < 0.001). IgG S1 antibody comparison between S1 immunized mice (red discontinuous line) vs. S1+N immunized mice (continuous purple line) (Figure 3E) showed differences that were statistically significant on day 80 (* *p* < 0.05) IgG RBD antibody comparison between RBD immunized mice (green discontinuous line) vs. RBD+N immunized mice (Continuous blue line) showed differences that were statistically significant in day 60 (** *p* < 0.01) and 80 (** *p* < 0.01) (Figure 3F). In summary, higher IgG antibody levels were recorded in sera from mice immunized with a combination of proteins (S1+N or RBD+N) compared to the groups of mice immunized with the recombinant proteins alone (N, S1, or RBD).

IgM or IgG antibodies against the three recombinant proteins were not detected in mice injected with adjuvant only. These results suggested that immunization with a combination of either S1 or RBD with the N protein induces a better IgG response.

### 3.4. Isotype Determination

When proteins were used for immunization, they elicit a Th2-associated, predominant IgG1 antibody response. The IgG1 subclass of antibodies were statistically more abundant than the IgG2a subclass antibodies (associated with the Th1 response) in all groups (** *p* < 0.01) (Figure 4C). IgG1 subclass antibodies against the RBD protein showed the highest concentration compared to antibodies against N or S1 proteins.

### 3.5. Determination of the Percentage of Neutralization

Having confirmed that the three antigens (S1, RBD, and N) can elicit a specific antibody response in the IgM and IgG classes, we further explored whether these antibodies could neutralize the virus, as cumulative data show that the protective immune response against COVID-19 is based on the presence of neutralizing antibodies [36]. The neutralizing potential of antibodies induced by recombinant proteins was evaluated using a simulated neutralization assay to avoid the need for high biocontainment laboratories (i.e., BSL-3). The simulated neutralization assay detects neutralizing antibodies against SARS-CoV-2 in serum by measuring the binding inhibition of biotinylated ACE2 to immobilized RBD [35]. Furthermore, the protein–protein interaction between RBD/ACE2 is inhibited if neutralizing antibodies against the RBD domain of SARS-CoV-2 are present in the test serum.

We observed neutralizing activity in the sera obtained from the S1, RBD, and RBD+N or S1+N immunized groups (Figure 5). No neutralizing activity was observed in groups immunized with N alone or treated with vehicle only. Notably, we observed a higher percentage of neutralizing activity in the groups immunized with a combination of proteins (S1+N and RBD+N) than in those immunized with S1 or RBD alone. Differences between immunized mice in combination with N protein vs. mice immunized just with S1 (*** *p* < 0.001) or RBD (*p* < 0.01) were statistically significant (Figure 5). These results suggest that the combination of N protein with other immunization antigens induces higher levels of neutralizing antibodies.

### 3.6. Cellular Response in Immunized Mice

Cumulative reports have shown that the N protein is highly immunogenic and is very well conserved among different coronaviruses, making it an attractive alternative in subunit protein vaccination [27]. Studies have shown that cytokines such as IFN-γ, IL-2, IL-4, and IL-5 produced by TCD4+ cells contribute to the maturation and differentiation of B cells [37]. Thus, we evaluated the cytokine response of total T cells extracted from the spleens of immunized mice before and after stimulation with the recombinant SARS-CoV-2 proteins. Splenocytes were stimulated with 2 μg of the N, S1, and RBD proteins for 48 h, and the levels of cytokines were detected using the Cytometric Bead Array (CBA) Mouse Th1/Th2 Cytokine kit. We observed the secretion of cytokines including TNF-α, IFN-γ, and IL-2 in response to stimulation with the recombinant SARS-CoV-2 proteins (Figure 6A–C). Differences between stimulated cells vs. unstimulated cells were statistically significant (*** *p* < 0.001). Notably, higher levels of these three cytokines were observed in the group of mice immunized with a combination of antigens (S1+N and RBD+N) than in response to only N protein stimulation. These results suggest that immunization with more than one antigen induces a robust cellular immune response.

### 3.7. Cross Reaction with RBD Variants

ELISA was used to determine the capacity of sera from immunized mice to recognize some VOCs. RBD based on sequences from the Wuhan, Alpha, and Beta SARS-CoV-2 variants were expressed in *E*. *coli*, purified, and used to coat ELISA plates. Figure 7 depicts the recognition of these RBD variants by sera from immunized mice. Specific IgG antibodies against the RBD Wuhan variant expressed in Expi293 were used as controls. IgG antibodies against RBD variants were identified in the RBD, RBD+N, S1, and S1+N groups. The highest levels of antibodies were observed in the RBD and RBD+N groups. A decrease in antibody levels was observed in response to RBD Alpha and Beta variants compared to the Wuhan RBD. However, altogether no significant differences were observed (*p* > 0.05) (Figure 7).

## 4. Discussion

Recently, a second-generation COVID-19 vaccine concept has emerged, searching for alternative antigens that can induce a more robust immune response, ideally even against the new emerging variants of SARS-CoV-2. In this study, we report improvement in the cellular and humoral immune response after immunization with a combination of two recombinant SARS-CoV-2 proteins (N+RBD and N+S1) in a mouse model.

The SARS-CoV-2 N protein has gained increasing interest as an alternative antigen that can induce a robust and long-lasting cellular immune response. Five vaccines currently in the clinical stage use the N protein as an antigen, and the other three use a combination of the N protein with the spike proteins [38]. The rationale is to have one antigen capable of inducing neutralizing antibodies (spike) and a second antigen (N) that confers T-cell-mediated immunity [39]. Previous work on SARS-CoV-infected individuals showed that a T-cell response against the N protein was observed 11 years after the initial infection [32]. In contrast, antibody response against the spike protein was maintained only three years after the infection was registered [40].

In previous studies, immunizations have been performed using bacterial proteins [41,42] or adenoviral vaccines [24,43]. Furthermore, it has been shown that the use of mammalian cell (human) expression systems produces high-quality recombinant proteins in terms of post-translation modifications, with structural and biochemical properties similar to the viral proteins produced during natural infection [44]. Currently, vaccines against influenza and hepatitis B, which are safely and effectively administered to the general population, are also based on recombinant proteins [45,46].

To date, fewer studies have analyzed protective immunity induced after immunization with a combination of SARS-CoV-2 antigens along with the N protein [24,25,26,41,47,48,49,50]. In this study, we show that recombinant N, S1, and RBD antigens of SARS-CoV-2 trigger specific IgM and IgG antibody responses. However, when the spike protein domains of S1 or RBD are combined with N, a much stronger cellular and humoral immune response is observed in the mouse model. This result seems to contradict the observations of Deming [51], who evaluated the immune response of mice immunized with SARS-CoV viral particles (VRPs) expressing both S and N proteins. In their report, the levels of IgG antibodies against protein S in mice immunized with VRPs expressing both SARS-CoV antigens (S and N) were not different from those of mice immunized with VRPs expressing only the S protein.

In this study, we observed that the antibodies induced by the three recombinant proteins were able to recognize the SARS-CoV-2 virus in the infected cells. As reported earlier, we observed that the N protein localized near the nuclei, and the spike proteins, were distributed across the cytoplasm [52,53]. These findings are important because they suggest that immunized mice may respond to native SARS-CoV-2 antigens upon infection.

After immunization with the recombinant proteins, we observed an elevation in both IgG1 and IgG2a subclasses’ mediated immune responses against the S and N proteins of SARS-CoV-2, with a predominance of the former, similar to other reported preclinical studies [24,41,54].

Neutralizing antibody activity is essential for resolving the SARS-CoV-2 infection [36]. Human studies have reported that a large proportion of neutralizing antibodies against SARS-CoV-2 are directed against the RBD [55]. Here, we used a simulated neutralization assay with immobilized RBD to determine the neutralizing capacity of the antibodies present in the sera of immunized mice to block the binding of the ACE2 receptor [35]. The sera of RBD-immunized mice showed a neutralization percentage of approximately 75%, closely followed by the sera of mice immunized with the recombinant protein S1, which contains the RBD domain. These findings align with the previous reports and suggest that the IgG1 subclass of antibodies, predominantly detected in the sera of immunized mice, play an important role in neutralizing the SARS-CoV-2 infection.

Considering that the N protein induces a cooperative T-cell response and that this response influences humoral immunity [56], we performed combined immunization with S1+N and RBD+N recombinant proteins. Neutralizing activity was observed in both groups. Notably, neutralization was higher in the mice immunized with the recombinant antigens than in mice immunized with single antigens. Rice et al. [24] used an adenoviral vector (Ad5) that simultaneously expressed the S protein and an N protein domain to immunize mice. They detected neutralizing IgG antibodies against the spike protein of SARS-CoV-2. In our study, the combination of the two recombinant proteins, RBD or S1, with the N protein resulted in a neutralizing activity of approximately 90%. This improvement in the immune response may be attributed to the cooperative T-cell response induced by the N protein via different cytokines. Hong and colleagues [41] reported that the combination of RBD-P2 and N recombinant proteins of SARS-CoV-2 induced an increased activation of CD4+ and CD8+ T cells in immunized mice. Moreover, in non-human primates, this combined immunization contributes to the elimination of the virus from the lungs. Evidence suggests that immunization with the N protein elicits a more robust immune response based on the stimulus that this protein exerts on CD4+ T cells. Zhao et al. [39] observed that immunization with a viral vector expressing a region of the N protein of SARS-CoV induced the production of IFN-γ-producing CD4+ T cells in mice. Moreover, these CD4+ cells offer protection against the SARS-CoV challenge. In 2020, Zhuang et al. [57] immunized mice expressing the human ACE-2 receptor with a viral vector containing a region of the N SARS-CoV-2 protein. They observed IFN-γ-producing T-cell induction and survival of animals upon SARS-CoV-2 infection. Furthermore, Dangi et al. [48] showed that combining a spike-based vaccine with a nucleocapsid-based vaccine provides protection to the lungs and the brain. This suggests that nucleocapsid-specific immunity enhances the distal control of SARS-CoV-2.

Regarding cellular immunity, we observed IFN-γ production in response to S1, RBD, and N protein stimulation of splenocytes from immunized mice. Higher levels of IFN-γ were recorded in groups immunized with the N protein, particularly in those with a combination of S1+N and RBD+N. IFN-γ is a cytokine involved in isotype switching from IgM to IgG antibodies and in affinity maturation [58]. We suggest that the activity of this cytokine could be responsible, at least in part, for the higher levels of IgG antibodies found in mice immunized with the protein mixture. IL-2 is an important cytokine for the maturation and survival of B-lymphocytes [59]. Similarly, we observed higher levels of IL-2 in groups immunized with the N protein combinations and hypothesized that this cytokine could also influence the antibody levels.

In 2022, Ghaemi et al. [60] showed that subcutaneous immunization of BALB/c female mice with the combination of RBD and N in a saponin formulation induced a strong immune response in comparison to immunization with RBD or N proteins alone. Higher levels of IgG antibodies, neutralization activity, and production of cytokines such as IFN-γ, IL-4, and IL-12 were found in groups immunized with the combination of proteins. The effect of administering S1+N and S2+N proteins with Allydrogel+MPLA has been evaluated by Özcengiz et al. [42] in a mouse model. They found higher levels of antibodies in mice groups immunized with combined N proteins. Similar results have been found in our research. However, one of the limitations of our study is that immunization was performed with Freud´s adjuvant, which is not approved for use in humans. We found a higher immune response with a combination of RBD+N and S1+N proteins. Unlike previous studies, we expressed proteins in eukaryotic systems. Therefore, they are post-transcriptionally modified and attain structural conformation similar to the native SARS-CoV-2 proteins. Altogether, the recombinant proteins generated in this study can be further used to evaluate other adjuvants, such as Allydrogel and MPLA, in the future.

In our study, the serum from mice immunized with proteins based on Wuhan sequences of SARS-CoV-2 was able to recognize alpha and beta VOCs. These findings support the cross-protection observed in vaccinated populations that are partially protected against VOCs such as alpha, beta, and delta. Our study reinforces the results in the report of Xu and Liu [61,62], who determined in preclinical tests that serum from animals immunized with Wuhan Spike or Wuhan RBD proteins had neutralizing activity against VOCs such as alpha, beta, and delta.

This study reports a strategy to combine the N protein with the S antigen of SARS-CoV-2 to induce an improved T-cell-mediated immune response that participates in protective non-neutralizing effector mechanisms. This combined antigen may be included in future vaccines.

One limitation of our study is that we had to use simulated neutralization tests to assay for neutralizing antibodies because we had limited access to a BSL Level 3 facility. Mice challenged with SARS-CoV-2 could not be used for the same reason. Nonetheless, the present study demonstrates that the combination of N and S recombinant proteins from SARS-CoV-2 expressed in a mammalian system elicits a specific cellular and humoral immune response and may be suitable as a potential vaccine candidate against SARS-CoV-2.

## Figures and Tables

**Figure 1 vaccines-11-00864-f001:**
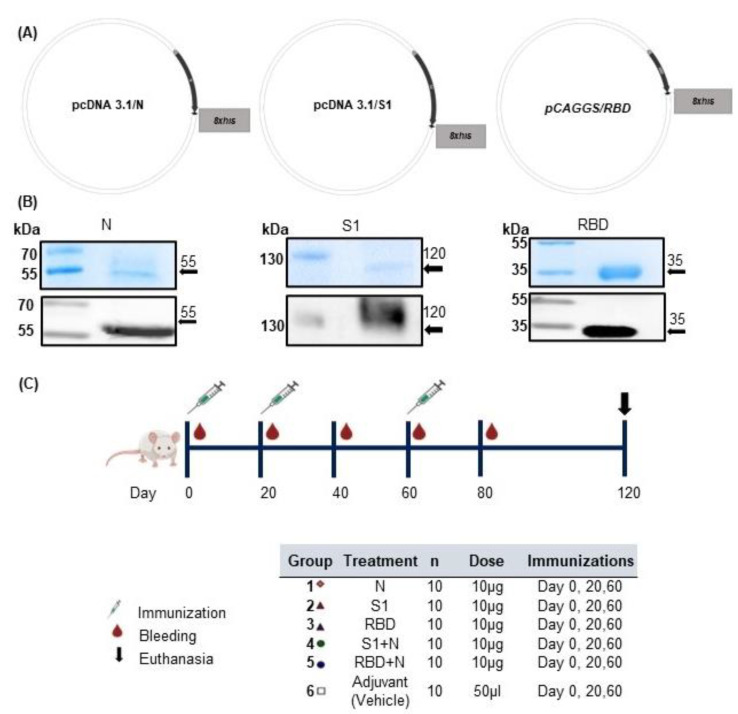
Design and expression of recombinant proteins and immunization scheme. (**A**) Schematic diagrams of plasmids, namely, pcDNA3.1/N, pcDNA3.1/S1, and pCAGGS/RBD. (**B**) SDS-PAGE and WB of the purified recombinant proteins N, S1, and RBD of SARS-CoV-2. (**C**) Scheme of immunization with recombinant proteins.

**Figure 2 vaccines-11-00864-f002:**
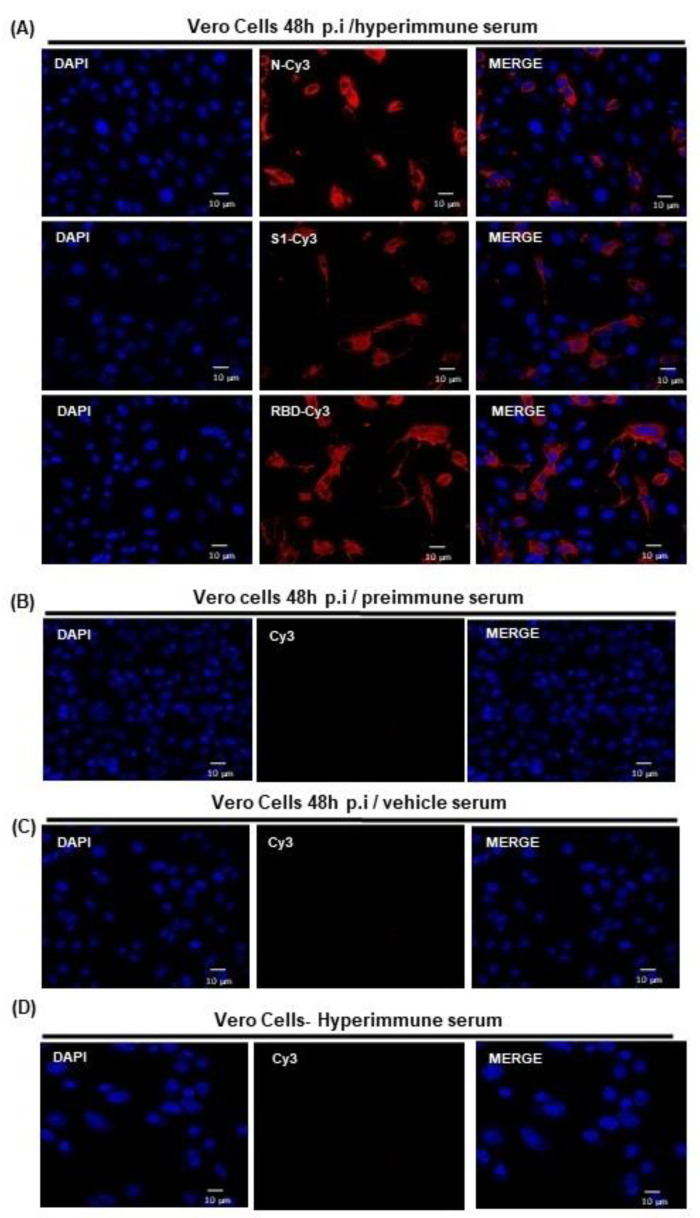
Immunofluorescence assays on SARS-CoV-2-infected Vero cells using sera from mice immunized with recombinant proteins. (**A**) SARS-CoV-2-infected Vero cells were exposed to pooled hyperimmune sera from mice immunized with SARS-CoV-2 recombinant proteins N, S1, and RBD, which were used as primary antibodies and visualized using a Cy3-conjugated goat anti-mouse IgG secondary antibody. Perinuclear and cytoplasmic distribution of N and of S1 and RBD, respectively, can be visualized. (**B**,**C**) Infected cells were analyzed using a pool of preimmune sera and sera from mice immunized with the vehicle alone. No recognition signal was observed. (**D**) Uninfected Vero cells incubated with hyperimmune sera also did not show localization of any of these proteins.

**Figure 3 vaccines-11-00864-f003:**
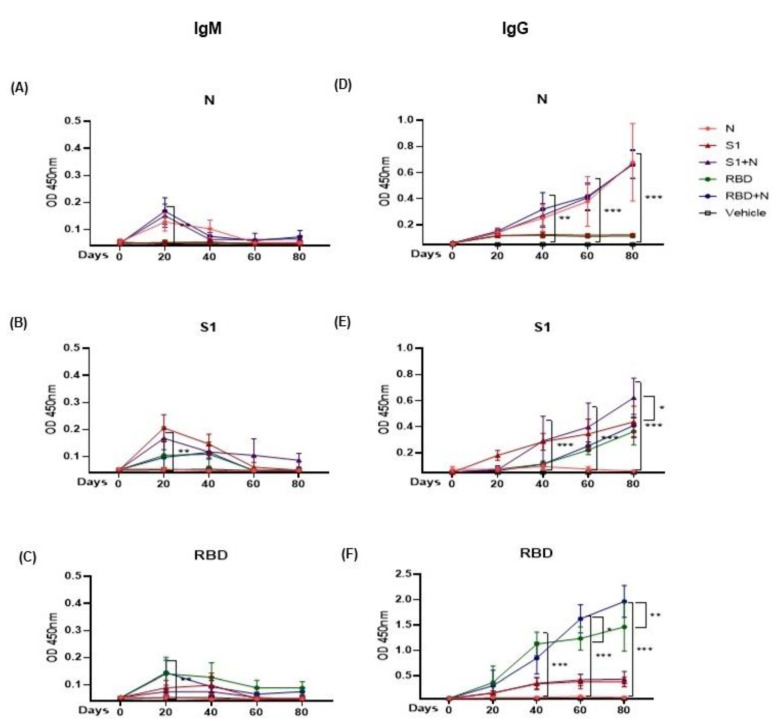
Immunization with recombinant proteins induces specific IgM (**A**–**C**) and IgG (**D**–**F**) anti-SARS-CoV-2 N, S1, and RBD antibodies, as evaluated using ELISAs. Sera were diluted to 1:200, and the results are expressed as O.D. at 450 nm, and the data of each group of 10 mice are plotted. For more than two samples, nonparametric ANOVA was used; bars represent the mean ± SD. * *p* < 0.05, ** *p* < 0.01, *** *p* < 0.001. *p* ≤ 0.05 was considered statistically significant. Prism v.8 software was used for statistical analyses (GraphPad Inc., La Jolla, CA, USA).

**Figure 4 vaccines-11-00864-f004:**
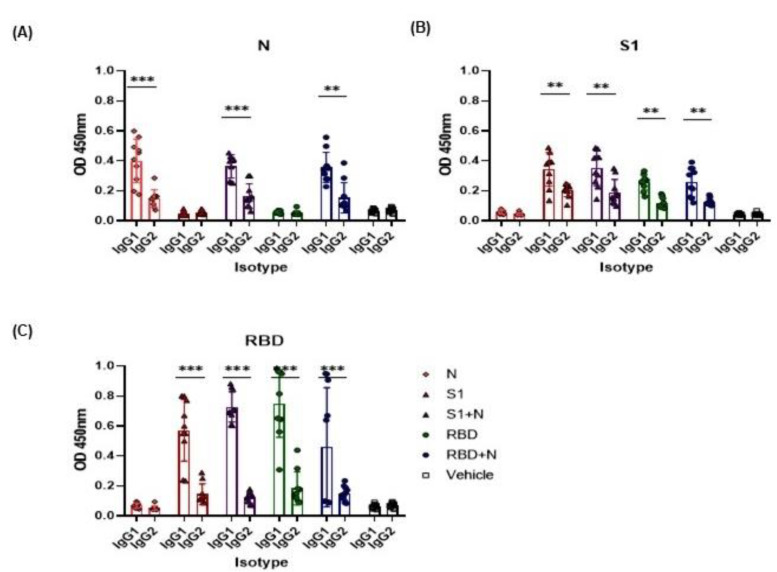
Recombinant proteins N (**A**), S1 (**B**), and RBD (**C**) of SARS-CoV-2 induce IgG1 and IgG2 antibody responses. The IgG subclass antibodies against the N, S1, and RBD proteins of SARS-CoV-2 were evaluated using ELISA 80 days after immunization. Sera were diluted to 1:100 and O.D. at 450 nm was measured. Student’s *t*-test was used for statistical analysis; bars represent the mean ± SD. ** *p* < 0.01, *** *p* < 0.001. *p* ≤ 0.05 was considered statistically significant. Prism v.8 software was used for statistical analyses (GraphPad Inc., La Jolla, CA, USA).

**Figure 5 vaccines-11-00864-f005:**
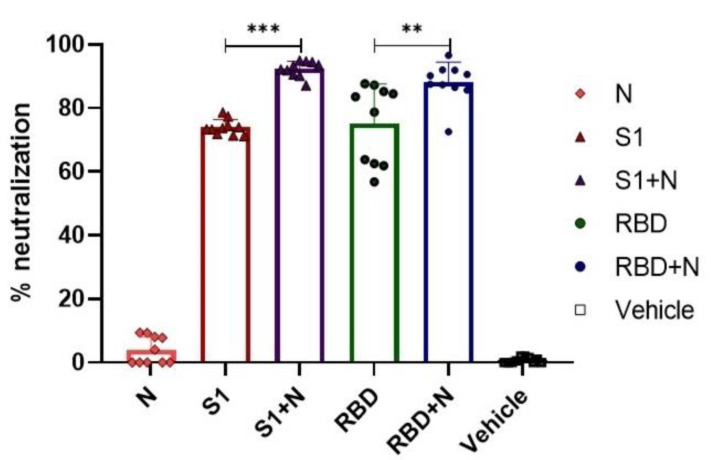
Combination with N protein results in higher levels of neutralizing antibodies in mice. The serum-neutralizing activity of immunized mice was evaluated using a simulated neutralization assay. Each bar represents the percentage of neutralizing activity observed in the groups immunized with indicated proteins. Sera were diluted 1:20. Student’s *t*-test was used; bars represent the mean ± SD. ** *p* < 0.01, *** *p* < 0.001. *p* ≤ 0.05 was considered statistically significant. Prism v.8 software was used for statistical analyses (GraphPad Inc., La Jolla, CA, USA).

**Figure 6 vaccines-11-00864-f006:**
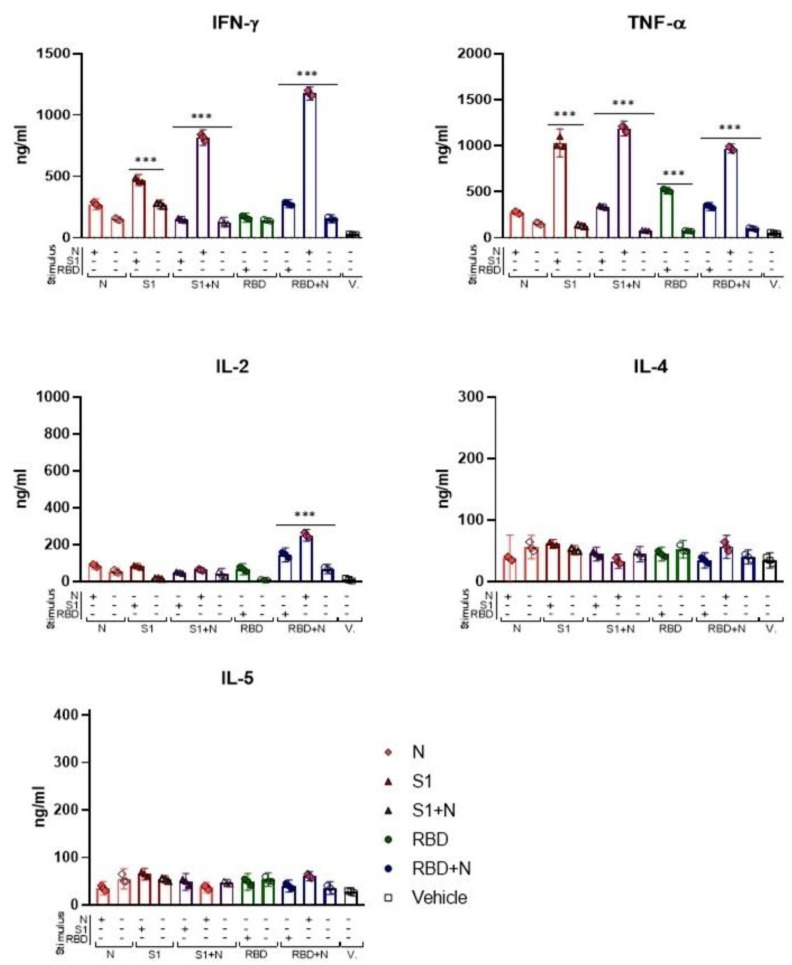
Immunization with recombinant proteins elicits a cellular response. Splenocytes from mice immunized with recombinant proteins alone or in combination were stimulated with 2 μg/mL of N, S1, and RBD recombinant proteins. No stimulated cells were used as controls. After 48 h, the levels of IFN-γ, TNF-α, IL-2, IL-4, and IL-5 were determined using CBAs in the supernatants of these cells. Each point represents a pool of three mice, and each bar represents an average of a group of 3 mice. A nonparametric ANOVA test was used; bars represent the mean ± SD. *** *p* < 0.001. *p* ≤ 0.05 was considered statistically significant. Prism v.8 software was used for statistical analyses (GraphPad Inc., La Jolla, CA, USA).

**Figure 7 vaccines-11-00864-f007:**
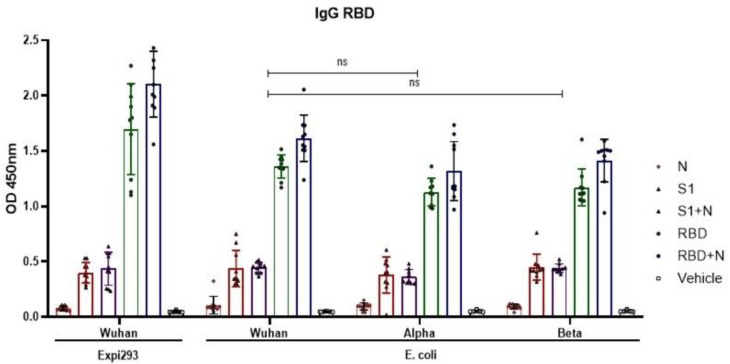
Recognition of RBD variants by sera from immunized mice. IgG antibodies against RBD Wuhan, Alpha, and Beta were evaluated using ELISA. Sera were diluted to 1:200. IgG antibodies were observed in response to RBD variants expressed in the *E. coli* system. Lower levels of these antibodies were detected in the variant groups than in the RBD Wuhan group, although the difference was not statistically significant. Bars represent the mean ± SD. *p* ≤ 0.05 was considered statistically significant. The Wilcoxon test in Prism v.8 software was used for statistical analyses (GraphPad Inc., La Jolla, CA, USA).

## Data Availability

The data presented in this study are available on request from the corresponding author.

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
