# Peer review of "Combination of Recombinant Proteins S1/N and RBD/N as Potential Vaccine Candidates"

_vaccines, 2023, doi:10.3390/vaccines11040864_

Round 1

Reviewer 1 Report

The article submitted by Mendoza-Ramírez et al to Vaccines journal explains the effectiveness of combined use of N with RBD or S for vaccine development. The article is relevant and informative. However, I would strongly suggest doing mice work like challenge experiments to know the efficacy of N and combined usage in mice.

The manuscript needs to improve in many ways. There are reports of using N protein as vaccine candidate and I would ask authors to mention how their work is different from theirs or do they add any new information to this. The result section mentions the statistical significance. But it is not mentioned to which groups are compared. The figures are fuzzy and pixelated. The language needs to be improved as well as there are lot of errors in the text.

Line 30: Spelling error- spike

Line 96: Spelling error- Wuhan

Line 103: Spelling error- Prokaryotic

Line 104: Strike off- as well

Line 141: Strike off- “For RBD Wuhan, Alpha and Beta variants, micro plates were 141 coated overnight at 4 °C with 2 μg of the respective proteins diluted in PBS 1X pH 7.4.”

Line 142: Rewrite the sentence_ For variants, micro plates were coated overnight at 4 °C with 2 μg of RBD from Wuhan isolate, Alpha strain and Beta strain diluted in PBS 1X pH 7.4.

Line 197 Figure 1 : Spelling error- adjuvant and immunizations

Line 213 Figure 2 : Spelling error- hyperimmune serum also mention uninfected in (D).

Line 222 Section 3.3: As I suggested for the figure 3 (Below), when you compare the data with the adjuvant group, include the p value information in the text too, so that the reader can see how much significant is. Rather saying it is significantly significant, if you have p values in the text it would be great (For eg, include 20dpi N/S1+N/RBD+ N vs Adjuvant p<0.01 after the sentence “The group of mice immunized with N….” ends).

Figure 3: I feel the bar graphs are not that clear and conclusive. The legends were not easy to identify. Since the antibody response is a time course event, line graphs would be better comparing with the adjuvant groups. Please denote the limit of detection in the graphs as dotted line.  O.D. should be written as OD.

Figure 3, 4, 5: Mention which groups are compared for statistical significance in the legends.

Line 253: For Th1 and Th2 responses, please write h in small letter.

Line 295: Rephrase the sentence “To address the activation of CD4+ T cells triggered by recombinant proteins through IFN-γ, cytokines such as IL-2, IL-4, and IL-5 produced by TCD4+ cells may contribute to the maturation and differentiation of B cells.[37]”.

Line 305: Spelling error RBD

Line 318: Rephrase the sentence “The capacity to recognize some VOCs in the sera of mice immunized with recombinant proteins alone or in combination was determined using ELISA.”

Line 325: Not sure the following sentence is relevant there. As shown in Figure 3, higher levels of IgG antibodies against RBD were observed in the group immunized with the N protein.

Line 341: SARS CoV-2

Author Response

Manuscript ID: vaccines-2270112
Title: “Combination of Recombinant Proteins S1/N and RBD/N as Potential
Vaccine Candidate”

We thank the reviewers for their insightful and constructive comments and invaluable suggestions. I apologize for all the mistakes. The manuscript has been checked in light of the comments and reviewed by a professional editing company (EDITAGE) for grammar and readability. I attach the issued certificate.

Revisor 1

The article submitted by Mendoza-Ramírez et al to Vaccines journal explains the effectiveness of combined use of N with RBD or S for vaccine development. The article is relevant and informative. However, I would strongly suggest doing mice work like challenge experiments to know the efficacy of N and combined usage in mice.

Thank you very much for your valuable comment. You are absolutely right, it would be ideal to carry out the challenge with the homologous virus, however we cannot perform challenge experiments for the lack lab BSL3. Although the BALB/c mice are an excellent model for preclinical assays. However, the BALB/c transgenic mice expressing hACE2 is the most appropriate for challenge experiments.

The manuscript needs to improve in many ways. There are reports of using N protein as vaccine candidate and I would ask authors to mention how their work is different from theirs or do they add any new information to this.

Thanks for your accurate comment. Indeed, as time passed, more reports have been published, where the N protein is used. However, in all of them proteins produced in bacteria have been used for immunization. Since Spike and Nucleocapsid show posttranscriptional modifications, it is very important expressed these antigens the most similar to native proteins of SARS CoV-2

The result section mentions the statistical significance. But it is not mentioned to which groups are compared. The figures are fuzzy and pixelated. The language needs to be improved as well as there are lot of errors in the text.

Now in the text the figures are explain with more detail, and the statistical significance is described.

In addition, the figures were modified with higher resolution.

Regarding the language, the manuscript has been checked and reviewed by a professional editing company.

 Line 30: Spelling error- spike

Done

Line 96: Spelling error- Wuhan

Corrected

Line 103: Spelling error- Prokaryotic

Corrected

Line 104: Strike off- as well

Corrected

Line 141: Strike off- “ For variants, micro plates were coated overnight at 4 °C with 2 µg RBD proteins based in Wuhan isolate, Alpha variant and Beta variant  diluted in PBS 1X pH 7.4. RBD expressed in Expi293 cells was used as a control.

The sentence has been modified.

Line 197 Figure 1 : Spelling error- adjuvant and immunizations

The mistakes were corrected in the  figure and the corrected is attached below the original

Line 213 Figure 2 : Spelling error- hyperimmune serum also mention uninfected in (D).

The figure was corrected and is attached below the original

Line 222 Section 3.3: As I suggested for the figure 3 (Below), when you compare the data with the adjuvant group, include the p value information in the text too, so that the reader can see how much significant is. Rather saying it is significantly significant, if you have p values in the text it would be great (For eg, include 20dpi N/S1+N/RBD+ N vs Adjuvant p<0.01 after the sentence “The group of mice immunized with N….” ends).

The p value has been introduced in the text, the text has been modified and described with more detail.

Figure 3: I feel the bar graphs are not that clear and conclusive. The legends were not easy to identify. Since the antibody response is a time course event, line graphs would be better comparing with the adjuvant groups. Please denote the limit of detection in the graphs as dotted line.  O.D. should be written as OD.

Thank you very much for your assertive comment, We have modify the graphs according to suggestion

Figure 3, 4, 5: Mention which groups are compared for statistical significance in the legends.

The figures has been modified

Line 253: For Th1 and Th2 responses, please write h in small letter.

Corrected

Line 295: Rephrase the sentence “To address the activation of CD4+ T cells triggered by recombinant proteins through IFN-γ, cytokines such as IL-2, IL-4, and IL-5 produced by TCD4+ cells may contribute to the maturation and differentiation of B cells.[37]”.

We apologize for explain incorrectly. The sentences has re-wrote, as follow:

Studies have shown that cytokines such as IFN-γ, IL-2, IL-4, and IL-5 produced by TCD4+ cells contribute to the maturation and differentiation of B cells [37]

Line 305: Spelling error RBD

corrected

Line 318: Rephrase the sentence “The capacity to recognize some VOCs in the sera of mice immunized with recombinant proteins alone or in combination was determined using ELISA.”

We apologize for the miswriting and now we have rephrased the sentence.

ELISA was used to determine capacity of sera from immunized mice to recognizing some VOC

Line 325: Not sure the following sentence is relevant there. As shown in Figure 3, higher levels of IgG antibodies against RBD were observed in the group immunized with the N protein.

We have eliminated the sentence.

Line 341: SARS CoV-2

corrected

Manuscript ID: vaccines-2270112
Title: “Combination of Recombinant Proteins S1/N and RBD/N as Potential
Vaccine Candidate”

We thank the reviewers for their insightful and constructive comments and invaluable suggestions. I apologize for all the mistakes. The manuscript has been checked in light of the comments and reviewed by a professional editing company (EDITAGE) for grammar and readability. I attach the issued certificate.

Revisor 1

The article submitted by Mendoza-Ramírez et al to Vaccines journal explains the effectiveness of combined use of N with RBD or S for vaccine development. The article is relevant and informative. However, I would strongly suggest doing mice work like challenge experiments to know the efficacy of N and combined usage in mice.

Thank you very much for your valuable comment. You are absolutely right, it would be ideal to carry out the challenge with the homologous virus, however we cannot perform challenge experiments for the lack lab BSL3. Although the BALB/c mice are an excellent model for preclinical assays. However, the BALB/c transgenic mice expressing hACE2 is the most appropriate for challenge experiments.

The manuscript needs to improve in many ways. There are reports of using N protein as vaccine candidate and I would ask authors to mention how their work is different from theirs or do they add any new information to this.

Thanks for your accurate comment. Indeed, as time passed, more reports have been published, where the N protein is used. However, in all of them proteins produced in bacteria have been used for immunization. Since Spike and Nucleocapsid show posttranscriptional modifications, it is very important expressed these antigens the most similar to native proteins of SARS CoV-2

The result section mentions the statistical significance. But it is not mentioned to which groups are compared. The figures are fuzzy and pixelated. The language needs to be improved as well as there are lot of errors in the text.

Now in the text the figures are explain with more detail, and the statistical significance is described.

In addition, the figures were modified with higher resolution.

Regarding the language, the manuscript has been checked and reviewed by a professional editing company.

 Line 30: Spelling error- spike

Done

Line 96: Spelling error- Wuhan

Corrected

Line 103: Spelling error- Prokaryotic

Corrected

Line 104: Strike off- as well

Corrected

Line 141: Strike off- “ For variants, micro plates were coated overnight at 4 °C with 2 µg RBD proteins based in Wuhan isolate, Alpha variant and Beta variant  diluted in PBS 1X pH 7.4. RBD expressed in Expi293 cells was used as a control.

The sentence has been modified.

Line 197 Figure 1 : Spelling error- adjuvant and immunizations

The mistakes were corrected in the  figure and the corrected is attached below the original

Line 213 Figure 2 : Spelling error- hyperimmune serum also mention uninfected in (D).

The figure was corrected and is attached below the original

Line 222 Section 3.3: As I suggested for the figure 3 (Below), when you compare the data with the adjuvant group, include the p value information in the text too, so that the reader can see how much significant is. Rather saying it is significantly significant, if you have p values in the text it would be great (For eg, include 20dpi N/S1+N/RBD+ N vs Adjuvant p<0.01 after the sentence “The group of mice immunized with N….” ends).

The p value has been introduced in the text, the text has been modified and described with more detail.

Figure 3: I feel the bar graphs are not that clear and conclusive. The legends were not easy to identify. Since the antibody response is a time course event, line graphs would be better comparing with the adjuvant groups. Please denote the limit of detection in the graphs as dotted line.  O.D. should be written as OD.

Thank you very much for your assertive comment, We have modify the graphs according to suggestion

Figure 3, 4, 5: Mention which groups are compared for statistical significance in the legends.

The figures has been modified

Line 253: For Th1 and Th2 responses, please write h in small letter.

Corrected

Line 295: Rephrase the sentence “To address the activation of CD4+ T cells triggered by recombinant proteins through IFN-γ, cytokines such as IL-2, IL-4, and IL-5 produced by TCD4+ cells may contribute to the maturation and differentiation of B cells.[37]”.

We apologize for explain incorrectly. The sentences has re-wrote, as follow:

Studies have shown that cytokines such as IFN-γ, IL-2, IL-4, and IL-5 produced by TCD4+ cells contribute to the maturation and differentiation of B cells [37]

Line 305: Spelling error RBD

corrected

Line 318: Rephrase the sentence “The capacity to recognize some VOCs in the sera of mice immunized with recombinant proteins alone or in combination was determined using ELISA.”

We apologize for the miswriting and now we have rephrased the sentence.

ELISA was used to determine capacity of sera from immunized mice to recognizing some VOC

Line 325: Not sure the following sentence is relevant there. As shown in Figure 3, higher levels of IgG antibodies against RBD were observed in the group immunized with the N protein.

We have eliminated the sentence.

Line 341: SARS CoV-2

corrected

Manuscript ID: vaccines-2270112
Title: “Combination of Recombinant Proteins S1/N and RBD/N as Potential
Vaccine Candidate”

We thank the reviewers for their insightful and constructive comments and invaluable suggestions. I apologize for all the mistakes. The manuscript has been checked in light of the comments and reviewed by a professional editing company (EDITAGE) for grammar and readability. I attach the issued certificate.

Revisor 1

The article submitted by Mendoza-Ramírez et al to Vaccines journal explains the effectiveness of combined use of N with RBD or S for vaccine development. The article is relevant and informative. However, I would strongly suggest doing mice work like challenge experiments to know the efficacy of N and combined usage in mice.

Thank you very much for your valuable comment. You are absolutely right, it would be ideal to carry out the challenge with the homologous virus, however we cannot perform challenge experiments for the lack lab BSL3. Although the BALB/c mice are an excellent model for preclinical assays. However, the BALB/c transgenic mice expressing hACE2 is the most appropriate for challenge experiments.

The manuscript needs to improve in many ways. There are reports of using N protein as vaccine candidate and I would ask authors to mention how their work is different from theirs or do they add any new information to this.

Thanks for your accurate comment. Indeed, as time passed, more reports have been published, where the N protein is used. However, in all of them proteins produced in bacteria have been used for immunization. Since Spike and Nucleocapsid show posttranscriptional modifications, it is very important expressed these antigens the most similar to native proteins of SARS CoV-2

The result section mentions the statistical significance. But it is not mentioned to which groups are compared. The figures are fuzzy and pixelated. The language needs to be improved as well as there are lot of errors in the text.

Now in the text the figures are explain with more detail, and the statistical significance is described.

In addition, the figures were modified with higher resolution.

Regarding the language, the manuscript has been checked and reviewed by a professional editing company.

 Line 30: Spelling error- spike

Done

Line 96: Spelling error- Wuhan

Corrected

Line 103: Spelling error- Prokaryotic

Corrected

Line 104: Strike off- as well

Corrected

Line 141: Strike off- “ For variants, micro plates were coated overnight at 4 °C with 2 µg RBD proteins based in Wuhan isolate, Alpha variant and Beta variant  diluted in PBS 1X pH 7.4. RBD expressed in Expi293 cells was used as a control.

The sentence has been modified.

Line 197 Figure 1 : Spelling error- adjuvant and immunizations

The mistakes were corrected in the  figure and the corrected is attached below the original

Line 213 Figure 2 : Spelling error- hyperimmune serum also mention uninfected in (D).

The figure was corrected and is attached below the original

Line 222 Section 3.3: As I suggested for the figure 3 (Below), when you compare the data with the adjuvant group, include the p value information in the text too, so that the reader can see how much significant is. Rather saying it is significantly significant, if you have p values in the text it would be great (For eg, include 20dpi N/S1+N/RBD+ N vs Adjuvant p<0.01 after the sentence “The group of mice immunized with N….” ends).

The p value has been introduced in the text, the text has been modified and described with more detail.

Figure 3: I feel the bar graphs are not that clear and conclusive. The legends were not easy to identify. Since the antibody response is a time course event, line graphs would be better comparing with the adjuvant groups. Please denote the limit of detection in the graphs as dotted line.  O.D. should be written as OD.

Thank you very much for your assertive comment, We have modify the graphs according to suggestion

Figure 3, 4, 5: Mention which groups are compared for statistical significance in the legends.

The figures has been modified

Line 253: For Th1 and Th2 responses, please write h in small letter.

Corrected

Line 295: Rephrase the sentence “To address the activation of CD4+ T cells triggered by recombinant proteins through IFN-γ, cytokines such as IL-2, IL-4, and IL-5 produced by TCD4+ cells may contribute to the maturation and differentiation of B cells.[37]”.

We apologize for explain incorrectly. The sentences has re-wrote, as follow:

Studies have shown that cytokines such as IFN-γ, IL-2, IL-4, and IL-5 produced by TCD4+ cells contribute to the maturation and differentiation of B cells [37]

Line 305: Spelling error RBD

corrected

Line 318: Rephrase the sentence “The capacity to recognize some VOCs in the sera of mice immunized with recombinant proteins alone or in combination was determined using ELISA.”

We apologize for the miswriting and now we have rephrased the sentence.

ELISA was used to determine capacity of sera from immunized mice to recognizing some VOC

Line 325: Not sure the following sentence is relevant there. As shown in Figure 3, higher levels of IgG antibodies against RBD were observed in the group immunized with the N protein.

We have eliminated the sentence.

Line 341: SARS CoV-2

corrected

Manuscript ID: vaccines-2270112
Title: “Combination of Recombinant Proteins S1/N and RBD/N as Potential
Vaccine Candidate”

We thank the reviewers for their insightful and constructive comments and invaluable suggestions. I apologize for all the mistakes. The manuscript has been checked in light of the comments and reviewed by a professional editing company (EDITAGE) for grammar and readability. I attach the issued certificate.

Revisor 1

The article submitted by Mendoza-Ramírez et al to Vaccines journal explains the effectiveness of combined use of N with RBD or S for vaccine development. The article is relevant and informative. However, I would strongly suggest doing mice work like challenge experiments to know the efficacy of N and combined usage in mice.

Thank you very much for your valuable comment. You are absolutely right, it would be ideal to carry out the challenge with the homologous virus, however we cannot perform challenge experiments for the lack lab BSL3. Although the BALB/c mice are an excellent model for preclinical assays. However, the BALB/c transgenic mice expressing hACE2 is the most appropriate for challenge experiments.

The manuscript needs to improve in many ways. There are reports of using N protein as vaccine candidate and I would ask authors to mention how their work is different from theirs or do they add any new information to this.

Thanks for your accurate comment. Indeed, as time passed, more reports have been published, where the N protein is used. However, in all of them proteins produced in bacteria have been used for immunization. Since Spike and Nucleocapsid show posttranscriptional modifications, it is very important expressed these antigens the most similar to native proteins of SARS CoV-2

The result section mentions the statistical significance. But it is not mentioned to which groups are compared. The figures are fuzzy and pixelated. The language needs to be improved as well as there are lot of errors in the text.

Now in the text the figures are explain with more detail, and the statistical significance is described.

In addition, the figures were modified with higher resolution.

Regarding the language, the manuscript has been checked and reviewed by a professional editing company.

 Line 30: Spelling error- spike

Done

Line 96: Spelling error- Wuhan

Corrected

Line 103: Spelling error- Prokaryotic

Corrected

Line 104: Strike off- as well

Corrected

Line 141: Strike off- “ For variants, micro plates were coated overnight at 4 °C with 2 µg RBD proteins based in Wuhan isolate, Alpha variant and Beta variant  diluted in PBS 1X pH 7.4. RBD expressed in Expi293 cells was used as a control.

The sentence has been modified.

Line 197 Figure 1 : Spelling error- adjuvant and immunizations

The mistakes were corrected in the  figure and the corrected is attached below the original

Line 213 Figure 2 : Spelling error- hyperimmune serum also mention uninfected in (D).

The figure was corrected and is attached below the original

Line 222 Section 3.3: As I suggested for the figure 3 (Below), when you compare the data with the adjuvant group, include the p value information in the text too, so that the reader can see how much significant is. Rather saying it is significantly significant, if you have p values in the text it would be great (For eg, include 20dpi N/S1+N/RBD+ N vs Adjuvant p<0.01 after the sentence “The group of mice immunized with N….” ends).

The p value has been introduced in the text, the text has been modified and described with more detail.

Figure 3: I feel the bar graphs are not that clear and conclusive. The legends were not easy to identify. Since the antibody response is a time course event, line graphs would be better comparing with the adjuvant groups. Please denote the limit of detection in the graphs as dotted line.  O.D. should be written as OD.

Thank you very much for your assertive comment, We have modify the graphs according to suggestion

Figure 3, 4, 5: Mention which groups are compared for statistical significance in the legends.

The figures has been modified

Line 253: For Th1 and Th2 responses, please write h in small letter.

Corrected

Line 295: Rephrase the sentence “To address the activation of CD4+ T cells triggered by recombinant proteins through IFN-γ, cytokines such as IL-2, IL-4, and IL-5 produced by TCD4+ cells may contribute to the maturation and differentiation of B cells.[37]”.

We apologize for explain incorrectly. The sentences has re-wrote, as follow:

Studies have shown that cytokines such as IFN-γ, IL-2, IL-4, and IL-5 produced by TCD4+ cells contribute to the maturation and differentiation of B cells [37]

Line 305: Spelling error RBD

corrected

Line 318: Rephrase the sentence “The capacity to recognize some VOCs in the sera of mice immunized with recombinant proteins alone or in combination was determined using ELISA.”

We apologize for the miswriting and now we have rephrased the sentence.

ELISA was used to determine capacity of sera from immunized mice to recognizing some VOC

Line 325: Not sure the following sentence is relevant there. As shown in Figure 3, higher levels of IgG antibodies against RBD were observed in the group immunized with the N protein.

We have eliminated the sentence.

Line 341: SARS CoV-2

corrected

Manuscript ID: vaccines-2270112
Title: “Combination of Recombinant Proteins S1/N and RBD/N as Potential
Vaccine Candidate”

We thank the reviewers for their insightful and constructive comments and invaluable suggestions. I apologize for all the mistakes. The manuscript has been checked in light of the comments and reviewed by a professional editing company (EDITAGE) for grammar and readability. I attach the issued certificate.

Revisor 1

The article submitted by Mendoza-Ramírez et al to Vaccines journal explains the effectiveness of combined use of N with RBD or S for vaccine development. The article is relevant and informative. However, I would strongly suggest doing mice work like challenge experiments to know the efficacy of N and combined usage in mice.

Thank you very much for your valuable comment. You are absolutely right, it would be ideal to carry out the challenge with the homologous virus, however we cannot perform challenge experiments for the lack lab BSL3. Although the BALB/c mice are an excellent model for preclinical assays. However, the BALB/c transgenic mice expressing hACE2 is the most appropriate for challenge experiments.

The manuscript needs to improve in many ways. There are reports of using N protein as vaccine candidate and I would ask authors to mention how their work is different from theirs or do they add any new information to this.

Thanks for your accurate comment. Indeed, as time passed, more reports have been published, where the N protein is used. However, in all of them proteins produced in bacteria have been used for immunization. Since Spike and Nucleocapsid show posttranscriptional modifications, it is very important expressed these antigens the most similar to native proteins of SARS CoV-2

The result section mentions the statistical significance. But it is not mentioned to which groups are compared. The figures are fuzzy and pixelated. The language needs to be improved as well as there are lot of errors in the text.

Now in the text the figures are explain with more detail, and the statistical significance is described.

In addition, the figures were modified with higher resolution.

Regarding the language, the manuscript has been checked and reviewed by a professional editing company.

 Line 30: Spelling error- spike

Done

Line 96: Spelling error- Wuhan

Corrected

Line 103: Spelling error- Prokaryotic

Corrected

Line 104: Strike off- as well

Corrected

Line 141: Strike off- “ For variants, micro plates were coated overnight at 4 °C with 2 µg RBD proteins based in Wuhan isolate, Alpha variant and Beta variant  diluted in PBS 1X pH 7.4. RBD expressed in Expi293 cells was used as a control.

The sentence has been modified.

Line 197 Figure 1 : Spelling error- adjuvant and immunizations

The mistakes were corrected in the  figure and the corrected is attached below the original

Line 213 Figure 2 : Spelling error- hyperimmune serum also mention uninfected in (D).

The figure was corrected and is attached below the original

Line 222 Section 3.3: As I suggested for the figure 3 (Below), when you compare the data with the adjuvant group, include the p value information in the text too, so that the reader can see how much significant is. Rather saying it is significantly significant, if you have p values in the text it would be great (For eg, include 20dpi N/S1+N/RBD+ N vs Adjuvant p<0.01 after the sentence “The group of mice immunized with N….” ends).

The p value has been introduced in the text, the text has been modified and described with more detail.

Figure 3: I feel the bar graphs are not that clear and conclusive. The legends were not easy to identify. Since the antibody response is a time course event, line graphs would be better comparing with the adjuvant groups. Please denote the limit of detection in the graphs as dotted line.  O.D. should be written as OD.

Thank you very much for your assertive comment, We have modify the graphs according to suggestion

Figure 3, 4, 5: Mention which groups are compared for statistical significance in the legends.

The figures has been modified

Line 253: For Th1 and Th2 responses, please write h in small letter.

Corrected

Line 295: Rephrase the sentence “To address the activation of CD4+ T cells triggered by recombinant proteins through IFN-γ, cytokines such as IL-2, IL-4, and IL-5 produced by TCD4+ cells may contribute to the maturation and differentiation of B cells.[37]”.

We apologize for explain incorrectly. The sentences has re-wrote, as follow:

Studies have shown that cytokines such as IFN-γ, IL-2, IL-4, and IL-5 produced by TCD4+ cells contribute to the maturation and differentiation of B cells [37]

Line 305: Spelling error RBD

corrected

Line 318: Rephrase the sentence “The capacity to recognize some VOCs in the sera of mice immunized with recombinant proteins alone or in combination was determined using ELISA.”

We apologize for the miswriting and now we have rephrased the sentence.

ELISA was used to determine capacity of sera from immunized mice to recognizing some VOC

Line 325: Not sure the following sentence is relevant there. As shown in Figure 3, higher levels of IgG antibodies against RBD were observed in the group immunized with the N protein.

We have eliminated the sentence.

Line 341: SARS CoV-2

corrected

Manuscript ID: vaccines-2270112
Title: “Combination of Recombinant Proteins S1/N and RBD/N as Potential
Vaccine Candidate”

We thank the reviewers for their insightful and constructive comments and invaluable suggestions. I apologize for all the mistakes. The manuscript has been checked in light of the comments and reviewed by a professional editing company (EDITAGE) for grammar and readability. I attach the issued certificate.

Revisor 1

The article submitted by Mendoza-Ramírez et al to Vaccines journal explains the effectiveness of combined use of N with RBD or S for vaccine development. The article is relevant and informative. However, I would strongly suggest doing mice work like challenge experiments to know the efficacy of N and combined usage in mice.

Thank you very much for your valuable comment. You are absolutely right, it would be ideal to carry out the challenge with the homologous virus, however we cannot perform challenge experiments for the lack lab BSL3. Although the BALB/c mice are an excellent model for preclinical assays. However, the BALB/c transgenic mice expressing hACE2 is the most appropriate for challenge experiments.

The manuscript needs to improve in many ways. There are reports of using N protein as vaccine candidate and I would ask authors to mention how their work is different from theirs or do they add any new information to this.

Thanks for your accurate comment. Indeed, as time passed, more reports have been published, where the N protein is used. However, in all of them proteins produced in bacteria have been used for immunization. Since Spike and Nucleocapsid show posttranscriptional modifications, it is very important expressed these antigens the most similar to native proteins of SARS CoV-2

The result section mentions the statistical significance. But it is not mentioned to which groups are compared. The figures are fuzzy and pixelated. The language needs to be improved as well as there are lot of errors in the text.

Now in the text the figures are explain with more detail, and the statistical significance is described.

In addition, the figures were modified with higher resolution.

Regarding the language, the manuscript has been checked and reviewed by a professional editing company.

 Line 30: Spelling error- spike

Done

Line 96: Spelling error- Wuhan

Corrected

Line 103: Spelling error- Prokaryotic

Corrected

Line 104: Strike off- as well

Corrected

Line 141: Strike off- “ For variants, micro plates were coated overnight at 4 °C with 2 µg RBD proteins based in Wuhan isolate, Alpha variant and Beta variant  diluted in PBS 1X pH 7.4. RBD expressed in Expi293 cells was used as a control.

The sentence has been modified.

Line 197 Figure 1 : Spelling error- adjuvant and immunizations

The mistakes were corrected in the  figure and the corrected is attached below the original

Line 213 Figure 2 : Spelling error- hyperimmune serum also mention uninfected in (D).

The figure was corrected and is attached below the original

Line 222 Section 3.3: As I suggested for the figure 3 (Below), when you compare the data with the adjuvant group, include the p value information in the text too, so that the reader can see how much significant is. Rather saying it is significantly significant, if you have p values in the text it would be great (For eg, include 20dpi N/S1+N/RBD+ N vs Adjuvant p<0.01 after the sentence “The group of mice immunized with N….” ends).

The p value has been introduced in the text, the text has been modified and described with more detail.

Figure 3: I feel the bar graphs are not that clear and conclusive. The legends were not easy to identify. Since the antibody response is a time course event, line graphs would be better comparing with the adjuvant groups. Please denote the limit of detection in the graphs as dotted line.  O.D. should be written as OD.

Thank you very much for your assertive comment, We have modify the graphs according to suggestion

Figure 3, 4, 5: Mention which groups are compared for statistical significance in the legends.

The figures has been modified

Line 253: For Th1 and Th2 responses, please write h in small letter.

Corrected

Line 295: Rephrase the sentence “To address the activation of CD4+ T cells triggered by recombinant proteins through IFN-γ, cytokines such as IL-2, IL-4, and IL-5 produced by TCD4+ cells may contribute to the maturation and differentiation of B cells.[37]”.

We apologize for explain incorrectly. The sentences has re-wrote, as follow:

Studies have shown that cytokines such as IFN-γ, IL-2, IL-4, and IL-5 produced by TCD4+ cells contribute to the maturation and differentiation of B cells [37]

Line 305: Spelling error RBD

corrected

Line 318: Rephrase the sentence “The capacity to recognize some VOCs in the sera of mice immunized with recombinant proteins alone or in combination was determined using ELISA.”

We apologize for the miswriting and now we have rephrased the sentence.

ELISA was used to determine capacity of sera from immunized mice to recognizing some VOC

Line 325: Not sure the following sentence is relevant there. As shown in Figure 3, higher levels of IgG antibodies against RBD were observed in the group immunized with the N protein.

We have eliminated the sentence.

Line 341: SARS CoV-2

corrected

Reviewer 2 Report

Dear Editor

I hare revised the work: Combination of Recombinant Proteins S1/N and RBD/N as Potential Vaccine Candidates. It is an interesting contribution, however, I have some suggestions:

1.-There is an interesting article that need to be include in the discussion: https://www.ncbi.nlm.nih.gov/pmc/articles/PMC9494508/

2.-Authors need to support and explain why N protein favored the humoral and cellular responses.

3.- There no a combination of S1 + RBD.

4.- As authors mention, the post-translational changes are very important, however, the authors do not characterize chemically neither peptides nor post-translational changes. It must be by proteomic studies under LC-MS.

5.- Authors need to consider the corresponding responses according to the equimolar doses, due to using same quantity (10 ug) for all proteins represent different quantities of them.

6.- To these proteins could be interesting perform 3D representations regarding to the proteins with and without post-translational changes that could help to the explanation about the biological responses.

7.- Authors must considerer one of the SAR-CoV-2 vaccines (protein based) used during the recently human vaccination to compare their results. 

Author Response

Manuscript ID: vaccines-2270112
Title: “Combination of Recombinant Proteins S1/N and RBD/N as Potential
Vaccine Candidate”

We thank the reviewers for their insightful and constructive comments and invaluable suggestions. I apologize for all the mistakes. The manuscript has been checked in light of the comments and reviewed by a professional editing company (EDITAGE) for grammar and readability. I attach the issued certificate.

Reviewer 2

Dear Editor

I hare revised the work: Combination of Recombinant Proteins S1/N and RBD/N as Potential Vaccine Candidates. It is an interesting contribution, however, I have some suggestions:

1.-There is an interesting article that need to be include in the discussion:

https://www.ncbi.nlm.nih.gov/pmc/articles/PMC9494508/

Thank you very much for your suggestion, It is definitely a reference that should be included in the paper.

Now we have included the reference in the discussion, as you kindly suggest us.

2.-Authors need to support and explain why N protein favored the humoral and cellular responses.

Nucleocapsid protein is able to elicit a T long lasting cellular immune  response during the natural infection with the possibility of induce broad protection. On other hand the induction of antibody protective response elicited by Spike protein at the end may give an amplifier effect due the two antigens. Recently Renee L. Hajnik et al (2022)  suggest that  after vaccination of  mRNA-N  and mRNA-S coimmunization may induce an immune environment that favors the generation of S-specific( Hajnik et al., Sci. Transl. Med. 14, eabq1945 (2022)). However more work to elucidate the mechanisms for S and N specific immunity collaboration remain to be performed. 

3.- There no a combination of S1 + RBD.

No combination have been performed because S1 subunit contain the RBD domain

4.- As authors mention, the post-translational changes are very important, however, the authors do not characterize chemically neither peptides nor post-translational changes. It must be by proteomic studies under LC-MS.

Thank you very much for your important and interesting comment.

Post transcriptional modifications have been reported for spike and N proteins. Spike have multiple glycosylation in RBD region. Bioinformatic and experimental studies report that N presents glycosylations, phosphorylations, acetylations and methylations.

Many reports has demonstrated the modification on the proteins,

Additionally, we provide a supplementary figure adding an analysis of modifications

5.- Authors need to consider the corresponding responses according to the equimolar doses, due to using same quantity (10 ug) for all proteins represent different quantities of them.

Thank you very much for your insightful comment. You are right, regarding the amount of total protein inoculated to each mouse, since in the combination of two proteins, we adjust to the net amount of protein, and we effectively inoculate half of each protein when we combine N+ RBD or N+S1,

however, we did not want to exceed the amount of antigen inoculated in single doses, trying to avoid high immunized doses in mice.

However, despite this, we were able to observe the enhancing effect exerted by the N protein.

6.- To these proteins could be interesting perform 3D representations regarding to the proteins with and without post-translational changes that could help to the explanation about the biological responses.

Structure of chimeric protein was predicted by AlphaFold server. Parameters as “novo” prediction and minimization of energy were applied. Visualization and analysis of structures were performed using Chimera 1.15 software.

MODEL 3D 

7.- Authors must considerer one of the SAR-CoV-2 vaccines (protein based) used during the recently human vaccination to compare their results. 

We totally agree with you comment, regarding to the SAR-CoV-2 vaccines (protein based) must be included, unfortunately, none of the vaccines that were used in Mexico were protein origin, so we have no way to get them. In Mexico RNA and Adenovirus recombinant vaccines for SARS CoV2 were used.

Round 2

Reviewer 2 Report

Dear Editor

I have revised the update version, however, I see that authors built the 3D structures, but these were not validate structurally (ej. https://saves.mbi.ucla.edu/) and nor include include into results and discussion. For example, they can discuss the proteins protein surfaces related to immunogenic regions reported elsewhere, etc.

I addition, it is correct about the reported data elsewhere about the the sequences and post-translational modifications for N and S proteins, but not for your proteins obtained genetically in your Laboratory, if you have previously reports, please include these or carry out the corresponding characterization from western blots, NMR, MS, etc.    

Author Response

I have revised the update version, however, I see that authors built the 3D structures, but these were not validate structurally (ej. https://saves.mbi.ucla.edu/) and nor include into results and discussion. For example, they can discuss the proteins protein surfaces related to immunogenic regions reported elsewhere, etc.

Thank you for your observations and suggestions to extend the validation of the model that we have included in the supplementary information.

According the suggestion to use the server that you kindly provided us (UCLA-DOE LAB), we found the next Overall Quality Factors, S1=77.21,N=80.83 and RBD= 72.77

Furthermore I consider that is important to comment that the Alpha fold yield a confidence scores per residual (pLDDT) between 0 and 100. For the model we report we have a very high (pLDDT > 90) of, confident (90 > pLDDT > 70) of, low (70 > pLDDT > 50) of, and very low (pLDDT < 50). Values of our structure are the followings, pLDDT S1>70  pLDDT N>80 and pLDDT RBD>80

I addition, it is correct about the reported data elsewhere about the the sequences and post-translational modifications for N and S proteins, but not for your proteins obtained genetically in your Laboratory, if you have previously reports, please include these or carry out the corresponding characterization from western blots, NMR, MS, etc.

Thank you very much for your valuable comments. In fact, several studies have been reported for proteins expressed in other systems. I definitely believe that it will be very useful to have these data obtained experimentally, unfortunately, we did not do performed the  characterization for this work, since the final objective of this study was to make an experimental analysis in terms of immunogenicity with the recombinant proteins that we designed and expressed in the cells of Expi293 cells transfected.

On other hand unfortunately, we  do not have the Nuclear Magnetic Resonance (NMR) and Mass Spectrometry (MS) equipment  and we do not have the possibility to contract the services

.